# Measurement report: Mobile measurements to estimate urban methane emissions in Tokyo

Taku Umezawa[1,2], Yukio Terao[1], Masahito Ueyama[3], Satoshi Kameyama[1], Mark Lunt[4], James Lawrence France[4,5]

[1]National Institute for Environmental Studies, Tsukuba, 305-8506, Japan
[2]Graduate School of Science, Tohoku University, Sendai, 980-8578, Japan
[3]Graduate School of Agriculture, Osaka Metropolitan University, Sakai, 599-8531, Japan
[4]Environmental Defense Fund, New York, NY 10010, USA
[5]Department of Earth Sciences, Royal Holloway University of London, Egham, TW20 0EX, United Kingdom

*Correspondence to*: Taku Umezawa (umezawa.taku@nies.go.jp)

**Abstract.** To investigate distributions and magnitudes of methane ($CH_4$) emissions in Tokyo, the world's largest megacity, a vehicle-based mobile measurement was set up and 3-week measurement campaign was conducted in September to October 2023. As part of the campaign, we conducted a control release experiment to link downwind excess $CH_4$ values to $CH_4$ emission rate at the source. The empirical equation derived from the experiment was significantly different from those reported by previous studies, suggesting the limitation of such enhancement-to-emission rate conversion, which is a source of large uncertainty in estimating urban $CH_4$ emissions based on street-level measurements. The uncertainty stems from different experiment settings and underlying assumptions (e.g., source distance and height) which do not always represent actual urban measurement environments. The mobile measurement campaign covered a large extent of the Greater Tokyo Area with total driving distance of over 2000 km. Locations of $CH_4$ enhancement were identified and $C_2H_6$-to-$CH_4$ enhancement ratios were determined for individual locations to categorize them into biogenic, fossil fuel and combustion $CH_4$ sources. Among a total of 565 locations inferred as $CH_4$ sources, 53 % and 42 % were considered as biogenic and fossil fuel origins, respectively, with the rest being minor contributions from combustion. Based on the statistics of measured $CH_4$ excesses, $CH_4$ emissions were estimated for the specific areas where relatively high measurement coverage was achieved. In the areas with biogenic facilities (landfill and wastewater treatment plants), our emission estimates are well correlated with local government reporting, indicating actual key contributions of the waste-sector facilities. On the other hand, in the residential areas, $CH_4$ emissions were predominantly of fossil-fuel origin, with a magnitude comparable to the area with waste facilities. However, such fossil-fuel emissions are not accounted for in local government reporting. This result highlights the need for improved accounting of urban fossil fuel-related emissions.

## 1 Introduction

Methane ($CH_4$) is a strong greenhouse gas whose atmospheric abundance has increased over the industrial era (e.g., Etheridge et al. 1998; Umezawa et al. 2022) including the recent decades (e.g., Lan et al. 2025; Umezawa et al. 2025a) due to enhanced

anthropogenic emissions (e.g., Chandra et al. 2021, 2024). Because of its relatively short lifetime in the atmosphere (about a decade), reduction of $CH_4$ emissions is effective to contribute to near-term mitigation of climate change towards the Paris Agreement goal (e.g., Collins et al. 2018). The Global Methane Pledge initiative further calls for emission reduction actions

in the anthropogenic $CH_4$ source sectors. Despite great efforts, current understanding of the $CH_4$ budget remains incomplete from global to national scales (e.g., Chandra et al. 2021, 2024; Jackson et al. 2024; Niwa et al. 2025; Janardanan et al. 2024). Furthermore, for taking mitigation actions, accurate estimates of facility-scale emissions are needed (Varon et al. 2019; Maasakkers et al. 2022).

$CH_4$ is emitted from various sources including wetlands, agriculture, fossil fuels and combustion. To attribute atmospheric

$CH_4$ variations to contributions from different sources, previous studies utilized simultaneous measurements of isotopes of $CH_4$ (e.g., Quay et al. 1999; Umezawa et al. 2012; Morimoto et al. 2017; Michel et al. 2024) and other gas tracers such as carbon monoxide (CO) and ethane ($C_2H_6$) (e.g., Xiao et al. 2004; Baker et al. 2012; Simpson et al. 2012). In particular, due to coincident emissions of $CH_4$ and $C_2H_6$ from fossil fuel sources with characteristic hydrocarbon composition (e.g., Schwietzke et al. 2014), it has been shown that $C_2H_6$ measurements are useful to evaluate $CH_4$ emissions from oil and gas sectors (e.g.,

Peischl et al. 2013; Yakovitch et al. 2015).

Cities are considered to be sources of $CH_4$ with emissions mainly from energy and waste sectors (e.g., Hopkins et al. 2016a; Takano and Ueyama, 2021). The former includes fugitive emissions associated with downstream oil and gas supply chain (refining, storage, distribution and consumption), and the latter landfills and wastewater treatment. The latter is of biogenic origin, produced from anaerobic decomposition. Although activity-based $CH_4$ emission estimates for urban areas have been

examined (Marcotullio et al. 2013; Crippa et al. 2021), observation-based methodologies for citywide verification have been still under development. Eddy covariance measurements provide accurate $CH_4$ fluxes for footprint areas at urban sites (e.g., Helfter et al. 2016; Takano and Ueyama, 2021), but, in many cases, they represent emissions from partial areas of a large city. Atmospheric transport modelling (including inverse analysis), combined with atmospheric mole fraction variations at upwind and downwind measurement sites, also provides quantification of $CH_4$ emissions for urban areas where atmospheric

measurements and corresponding emission inventory data are available (McKain et al. 2015; Sargent et al. 2021; Saboya et al. 2022).

An increasing number of studies have examined on-street measurements using a vehicle to locate and quantify $CH_4$ emissions in cities (Zazzeri et al. 2015; Hopkins et al. 2016b; von Fischer et al. 2017; Weller et al. 2018; Maazallahi et al. 2020; Ars et al. 2020; Xueref-Remy et al. 2020; Defratyka et al. 2021; Fernandez et al. 2022; Wietzel and Schimidt, 2023; Joo et al. 2024;

Ueyama et al. 2025). These studies have shown characteristics of $CH_4$ emissions from worldwide large cities such as London (Zazzeri et al. 2015), Los Angels (Hopkins et al. 2016b), Tronto (Ars et al. 2020), Paris (Xueref-Remy et al. 2020; Defratyka et al. 2021), Seoul (Joo et al. 2024) and Osaka (Ueyama et al. 2025), which indicated strong emissions from natural gas distribution and waste management sectors in these urban areas. Combining vehicle-based measurement data from different cities, Vogel et al. (2024) presented an analysis of natural gas leakage in Canada and European countries. Recently, Ueyama

et al. (2025) conducted a vehicle-based $CH_4$ survey in Osaka, the second largest city in Japan. As part of the joint project of

Ueyama et al. (2025), this study presents analogous measurements in Tokyo, which is Japan's and currently the world's largest megacity in population (United Nations, 2019).

According to Tokyo Metropolitan Government, $CH_4$ emissions from Tokyo Metropolis were 21 kt $CH_4$ $yr^{-1}$ for 2023, which were predominantly from waste sectors (96.0 %) with minor contributions of fuel combustion (3.3 %) and agriculture (0.8 %) (Bureau of Environment of Tokyo Metropolitan Government, 2025). When aggregated for Tokyo Metropolis, the EDGAR (Emission Database for Global Atmospheric Research) dataset shows $CH_4$ emissions of 18 kt $CH_4$ $yr^{-1}$ for 2023, where waste, energy (including fossil fuel), and agriculture sectors constitute 45 %, 41 % and 14 % of the total emissions, respectively (Crippa et al. 2024). Given the general small decreasing trend of $CH_4$ emissions of Tokyo (e.g., 0.5 % $yr^{-1}$ in the Tokyo Metropolitan Government reporting), comparison of these datasets indicates discrepancy in magnitude and attribution of $CH_4$ sources in Tokyo between the local government reporting and the global data commonly used in the atmospheric science community, highlighting the importance to improve activity-based $CH_4$ emission datasets for Tokyo. In this study, we present our vehicle-based $CH_4$ measurements in Tokyo (including instrument evaluations), data analyses to identify $CH_4$ source locations and types, and current-best approximations of $CH_4$ emissions for specific areas where the data allows.

## 2 Method

### 2.1 Target area and measurement campaign

The Greater Tokyo Area is the world's most populated metropolitan area. According to the Japan Statistics Bureau, the Kanto Major Metropolitan Area, one of various definitions of the Greater Tokyo Area, is defined as the area that consists of all municipalities that have >1.5 % of their population (aged 15 and above) commuting to designated cities (Chiba, Kawasaki, Sagamihara, Saitama and Yokohama) or the 23 special wards of Tokyo Metropolis. The area's population was about 38 million in 2020, according to the Census. To our knowledge, no vehicle-based measurements of $CH_4$ and $C_2H_6$ have been ever reported within this area. In designing driving routes, we therefore prioritized coverage of large extent of the Greater Tokyo Area. In addition, particular focus was the area around the Yoyogi site (35.66° N, 139.68° E). Atmospheric $CH_4$ mole fraction measurement (not shown in this study) is ongoing at the site located in the central part of Tokyo, with the surrounding main land cover being residential buildings (Ishidoya et al. 2020; Sugawara et al. 2021). The Tokyo Bay area was also of interest, as part of the area is occupied by Japan's two major industrial zones (Keihin and Keiyo Industrial Zones) that hold heavy industries (e.g., steel mills, oil refineries, chemical plants and electricity generation). In the area, there is also Tokyo Bay-side Landfill in the Port of Tokyo, which is the final disposal site of solid wastes from the 23 special ward areas of Tokyo. In addition, the 23 special wards have 13 wastewater treatment plants, and they were also targets of the present measurement surveys. $CH_4$ emissions from the individual landfill and wastewater facilities are however beyond the scope of this study and will be investigated in a separate paper.

Our vehicle-based mobile measurements were conducted during daytime on 15 days (18–22 September, and 2–6, 16 and 18–21 October 2023: 14 weekdays and 1 Saturday). The measurement survey covered populated areas of Tokyo Metropolis as well as areas around the Tokyo Bay in Chiba and Kanagawa Prefectures, as shown in Fig. 1. The measurements were made on public roads only. The driving distance on each measurement day ranged from 71 to 208 km, depending on focus areas, and the total distance was 2012 km.

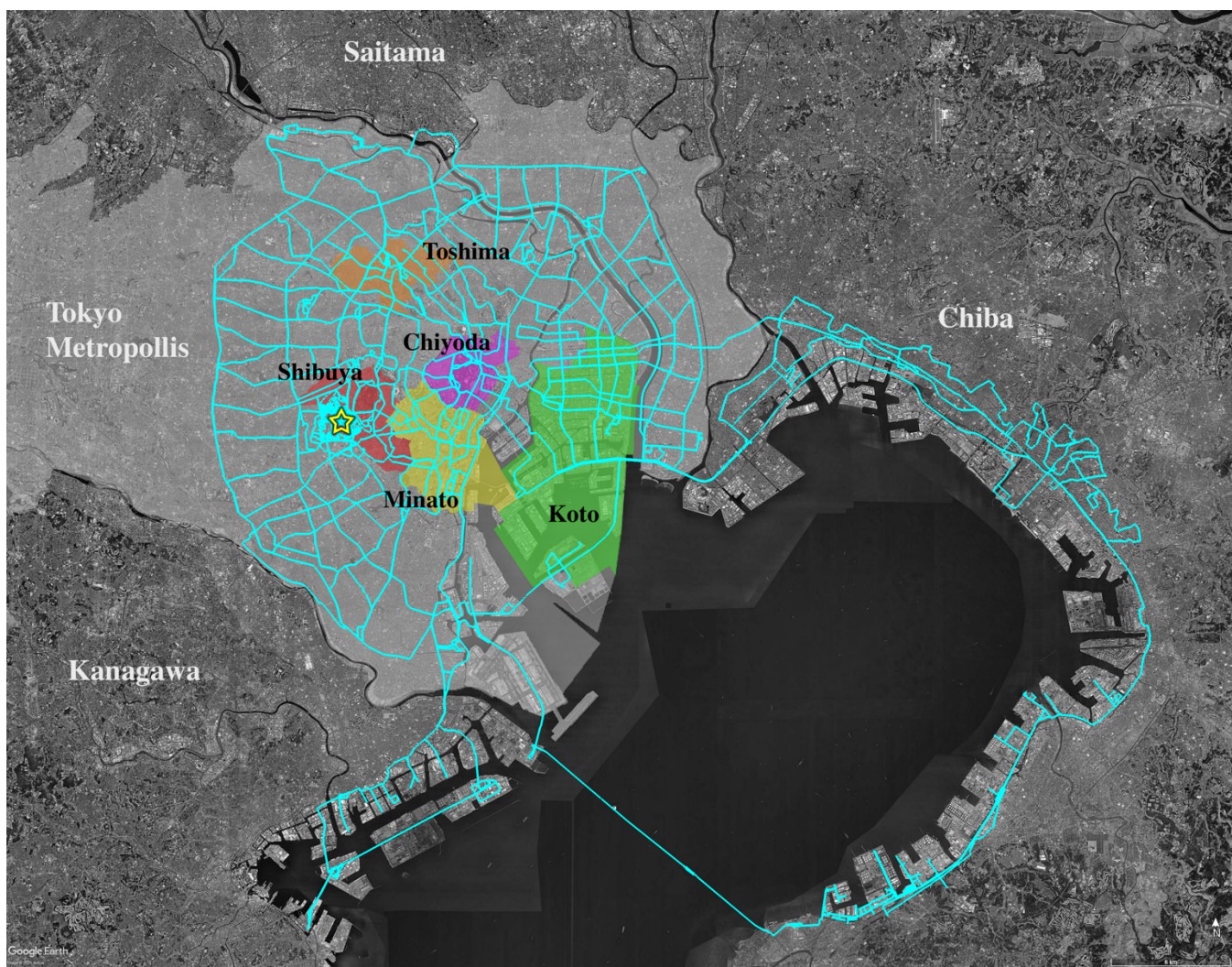

**Figure 1: A map showing the Greater Tokyo Area and the driving routes (light blue lines) of the measurement campaign conducted in September–October 2023. Tokyo Metropolis is shaded white and five selected special wards of Tokyo Metropolis (Chiyoda, Minato, Koto, Shibuya and Toshima) are shaded colors. A star in Shibuya Ward indicates the location of the site Yoyogi. The map imagery was from © Google Earth.**

## 2.2 Laboratory evaluation

To measure atmospheric mole fractions of $CH_4$ and $C_2H_6$, we used a MIRA Ultra gas analyzer (Aeris Technologies, USA).
The analyzer is a mid-infrared absorption spectrometer with a multi-pass cell (60 $cm^3$ volume). The cell pressure is maintained at ~240 mbar by an internal pump. More descriptions on the analyzer were previously given in Travis et al. (2020) and Commane et al. (2023).

To determine the $CH_4$ and $C_2H_6$ mole fractions in sample air, we prepared a suite of dry-air standard gases containing $CH_4$ and $C_2H_6$ at ambient to urban elevated mole fraction levels, ranging approximately from 2 to 5 µmol $mol^{-1}$ (hereafter denoted as
ppm) for $CH_4$ and 1 to 10 nmol $mol^{-1}$ (hereafter denoted as ppb) for $C_2H_6$. These standard gases were produced gravimetrically in 9.4 L aluminum cylinders by Taiyo Nippon Sanso JFP Corporation. The $CH_4$ and $C_2H_6$ mole fractions reported in this study are traceable to those in these standard gases. Prior to the measurements, the $CH_4$ mole fractions in the standard gases were measured by a cavity ring-down spectroscopy analyzer (G-2401, Picarro Inc., USA). The measurements confirmed that the $CH_4$ mole fractions in the standard gases show excellent linearity with reproducibility of <1.0 ppb. The standard gases were
also measured for $C_2H_6$ by a preconcentration and refocusing gas-chromatography mass-spectrometry (GC-MS) system (Umezawa et al. 2025b). The measurement results showed a good linearity of the standard gases with reproducibility of <0.1 ppb.

The repeatability of the MIRA Ultra analyzer has been evaluated by Allan-Werle variance plots (e.g., Commane et al. 2023). We measured compressed air in a 47 L cylinder for about 24 hours and calculated the Allan-Werle variance (not shown). The
same experiment was made twice: 6–7 September 2023 and 21–22 February 2024. The first experiment was soon after delivery of the analyzer and the second was made after the modified version of the software (data acquisition frequency increased) being implemented. Repeatability of MIRA Ultra for 100 s was 0.02 ppb for $CH_4$ and 0.004 ppb for $C_2H_6$, which were later improved to 0.007 ppb for $CH_4$ and 0.003 ppb for $C_2H_6$. These values exceed those reported by Commane et al. (2023) for the same product.

The standard gases were measured periodically to evaluate longer-term stability of the analyzer. The instrument showed good stability during the mobile measurement campaign period in Tokyo (18 September–21 October 2023), with the measured $CH_4$ values of the standard gases agreed to the nominal value within ~5 ppb at baseline level (~2 ppm). In contrast, the measured values for the standard gases with higher $CH_4$ mole fractions (up to ~5 ppm) showed differences up to ~25 ppb. As this possible bias (~0.025 ppm) is relatively small in comparison to the observed variability (>1 ppm excess values with respect to the
baseline), we apply no corrections for $CH_4$ measurements during the Tokyo mobile campaign measurements. It was also shown that measured $C_2H_6$ values by MIRA Ultra varied significantly from day to day. This suggests that raw $C_2H_6$ values reported from the analyzer are not compatible among different measurement days. We however confirmed that the measurement span covered by the standard gases remained constant (for the mole fraction range approximately from 1 to 10 ppb) over the course of measurements, indicating that the excess values in the $C_2H_6$ mole fraction were robust. We therefore report excess values
only for $C_2H_6$.

We evaluated two MIRA Ultra analyzers of two institutions (National Institute for Environmental Studies, NIES and Osaka Metropolitan University, OMU) for sensitivity to humidity as previously reported by Commane et al. (2023). The NIES and OMU analyzers were operated for the measurements in Tokyo (this study) and Osaka (Ueyama et al. 2025), respectively. Figure 2 displays the $CH_4$ and $C_2H_6$ mole fraction dependence on the amount of water vapor. The mole fractions here are reported values by the instrument and applied no corrections. In this experiment, a compressed dry air was introduced into the analyzer. The air passed through a moisture exchanger (ME-110-72COMP-4, Perma Pure LLC) housed in a food container in which a humidor pack (84 % RH, Boveda Inc.) was placed. Warm up of the container increased humidity inside the container, which also increased the amount of water vapor in the sample flow of the air via the moisture exchanger. To examine response with low humidity, silica gel was placed in the container instead of the humidor pack to dry the sample air.

As shown in Fig. 2, $CH_4$ measurement by the analyzer showed a quadratic curve (grey and black lines) expressed by:

$$\text{NIES: } Y = -0.05478 + 8.4828 \times 10^{-6}X - 3.2856 \times 10^{-10}X^2 \tag{1}$$

$$\text{OMU: } Y = -0.020178 + 4.0952 \times 10^{-6}X - 2.07786 \times 10^{-10}X^2 \tag{2}$$

Commane et al. (2023) reported the humidity response of the MIRA Ultra analyzer, but interestingly, their experiment indicated a different curve in shape. They reported that the $CH_4$ value from the analyzer showed a quadratic but monotonically decreased curve with increasing humidity. In contrast, our results in Fig. 2 suggest that the measured $CH_4$ value has a plateau at water vapor range of approximately 5000–15000 ppm. For the NIES MIRA Ultra, in the water vapor range of 7500–18000 ppm, the measured $CH_4$ value falls within −0.01 ppm relative to the top of the quadratic convex curve positioned at ~13000 ppm water vapor. This implies that measurements with water vapor in the above target range would report $CH_4$ values with better reproducibility. We therefore consider that (1) calibration of the analyzer should be made by reference gases humidified to as close as the target range, and (2) $CH_4$ values reported by the analyzer should be corrected according to the water vapor values.

For $C_2H_6$, the measured value from the NIES MIRA Ultra showed a slight increase with increasing humidity expressed by:

$$\text{NIES: } Y = -0.6521 + 4.8243 \times 10^{-5}X \tag{3}$$

In the above water vapor range (7500–18000 ppm), the $C_2H_6$ values are expected to fall within ±0.3 ppb from the nominal value. The $C_2H_6$ values reported by the analyzer should be corrected accordingly. In contrast, the MIRA Ultra analyzer of OMU did not show a consistent trend between experiments when the water vapor was increased and decreased (Fig. 2, bottom panel). Note that measurements by OMU MIRA Ultra are not included in this study and presented in the companion paper (Ueyama et al. 2025). It should be noted that the water response curve, as presented in this study, is likely different from instrument to instrument even within the same model product MIRA Ultra; we therefore recommend careful evaluation of each analyzer for high-accuracy measurements.

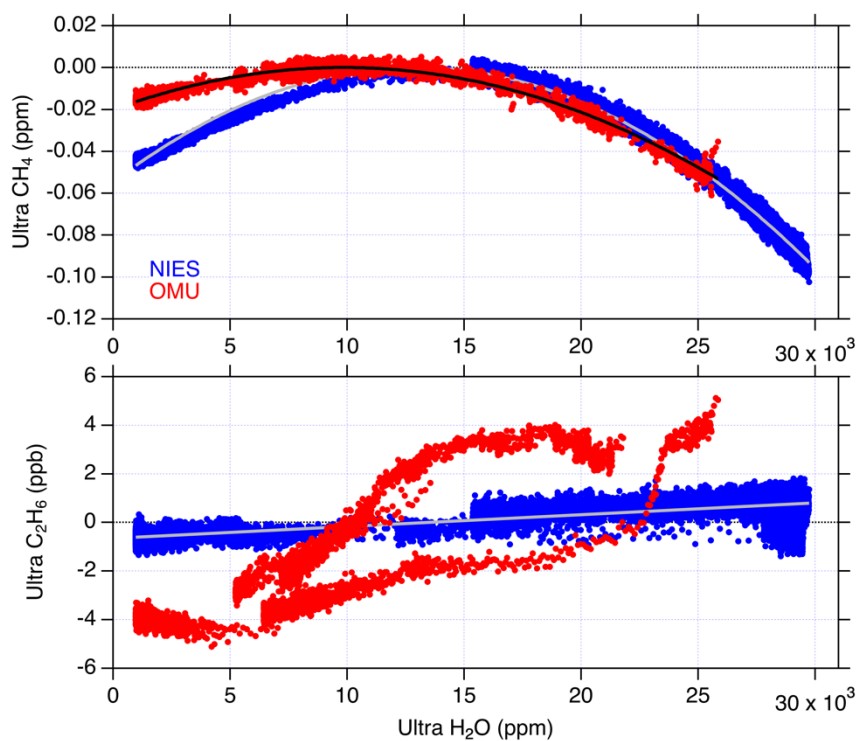

**Figure 2: Sensitivity of the CH₄ (top) and C₂H₆ (bottom) mole fractions to the water vapor of the two MIRA Ultra analyzer of NIES (blue) and OMU (red). Note that CH₄ and C₂H₆ mole fractions are plotted as differences from nominal values.**

### 2.3 Mobile measurement set-up

The vehicle-based measurement system consisted of the MIRA Ultra analyzer, a sample air inlet and connecting tubings, a GPS (Global Positioning System) receiver (16X-HVS, Garmin, USA), an anemometer (Portable Mini, Calypso Instruments, USA), a data logger (CR-1000X, Campbell Scientific, Inc., USA), and power supply. The car is equipped with the anemometer on the roof, the GPS receiver at the inner side of the front window, and the air inlet at front low part of the car (~0.5 m height from the ground). Sample air is drawn by the internal pump of the MIRA Ultra instrument. Length of the tubing from the air inlet to MIRA Ultra is 6.25 m (5-m 1/8" OD and 1.25-m 1/4" OD). Power is supplied from the car power socket (+12 VDC) via a power inverter to MIRA Ultra and the data acquisition system. The main component of the data acquisition system is the data logger, which collects data from the anemometer, GPS and MIRA Ultra. Prior to measurement, a gas mixture that contains 1 % of hydrocarbons including CH₄ and C₂H₆ was sprayed to the air inlet to measure response time from the air inlet to MIRA Ultra. The response time was 8 sec in most days with only a few days of 7 sec or 9 sec. The data were located with the response time corrected for each measurement day.

## 3 Control release experiment

Mobile vehicle-based measurements provide $CH_4$ mole fractions along the driving track so that emissions are indicated by enhancement from the background value. To give a preliminary estimate of $CH_4$ emission rate simply based on enhancements of the $CH_4$ mole fraction, we conducted a controlled $CH_4$ emission experiment. In the experiment, $CH_4$ outflow from a 47 L cylinder is controlled at a fixed flow rate for ~10 min, and the $CH_4$ gas is released from the downstream pipe at ~5 m height from the ground. The controlled flow rate ranged from 1.0 to 15.0 NL min$^{-1}$ at about 5-min intervals. The $CH_4$ mole fraction was measured ~50 m downwind by the MIRA Pico analyzer. The outlet pipe was visible from the measurement site and there was almost no obstacle that interfered wind advection in between except a mesh fence and bushes. The analyzer was subsequently calibrated by using a set of reference gases. Note that this experiment was not carried out by the MIRA Ultra analyzer, which was used in the mobile measurement campaign, but we confirmed consistency of response outputs from both analyzers within the measurement uncertainties.

The experiment was conducted on 28 July 2023. The first set of the experiment with the above varying flow rates started around 10:00 and ended around 12:30 local time, and it was repeated in the afternoon approximately from 14:00 to 16:30. The weather was sunny with temperatures of 31°C in the morning and up to 34°C in the afternoon. The easterly wind (direction of the outlet pipe to the measurement site) prevailed during the experiment with wind speed below 3 m s$^{-1}$. These weather conditions agree to those typically observed at the site and in Tokyo (Japan Meteorological Agency, 2025). From time series of observed $CH_4$ enhancements, the excess $CH_4$ was defined as the $CH_4$ mole fraction difference from a background value, which was calculated as the median values in the 3-min time window (±1.5 min from the individual data points). When the excess $CH_4$ exceeded +0.05 ppm from the background, the data point was tagged. Among the tagged data points with enhanced $CH_4$, the maximum excess $CH_4$ value in each emission period was plotted as a function of the $CH_4$ emission rate (Fig. 3a). The linear fit yielded the following relationship between the excess $CH_4$ (ppm) and $CH_4$ emission rate (g $CH_4$ min$^{-1}$):

$$\text{max excess } CH_4 = (0.118\pm0.117) + (0.033\pm0.019)\times\text{emission rate} \tag{4}$$

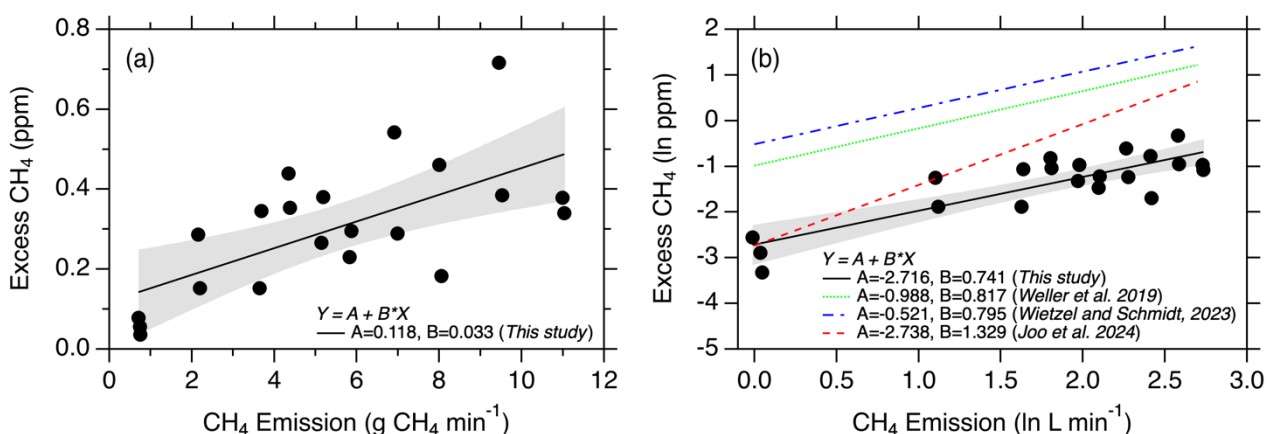

**Figure 3: (a) The observed maximum excess CH$_4$ as a function of the CH$_4$ emission rate. The original CH$_4$ emission rate in unit NL min$^{-1}$ was converted to g CH$_4$ min$^{-1}$ with the gas density of CH$_4$ (0.717 kg m$^{-3}$). (b) Same as (a), but both axes are on _ln_ scales. Our experiment data and the linear fit are shown in black and the linear relationships suggested by Weller et al. (2019), Wietzel and Schmidt (2023) and Joo et al. (2024) are shown by green dotted, blue dash-dotted and red dashed lines, respectively. The linear regression coefficients are also shown. The grey shades indicate the 95 % confidence bands.**

In Fig. 3b, the above linear relationship was converted into the natural log (_ln_) scales for comparison to those reported by previous studies (Weller et al. 2019; Wietzel and Schmidt, 2023; Joo et al. 2024). Our data aligned in the range of lower excess CH$_4$ with the slope being smaller than other studies, showing that our equation gives a larger estimate of the emission rate when given a same excess CH$_4$ value. The discrepancy may be attributable to difference in the experiment settings. Weller et al. (2019) and Wietzel and Schmidt (2023) assumed an average approximate distance between leak and measurement points to be 15.75 m and 7 m, respectively. Joo et al. (2024) assumed 0 m distance between leak and measurement location, as their test experiment was conducted on a narrow driving path surrounded by buildings. In contrast, our experiment data was collected with source distance of ~50 m after horizontal advection. These differences imply that distance to the source is an important factor. For the present experiment with distance of ~50 m, the equation by Weller et al. (2019) and Wietzel and Schmidt (2023) could underestimate the emission rate. Comparison to Joo et al. (2024) also suggests that the slope could be considerably different depending on surrounding environment of the experiments. Furthermore, in contrast to the 5-m height of CH$_4$ release in this study, CH$_4$ was released from the ground in the above three studies, where detection of leakage from underground pipes was assumed. The emission height is also one of decisive initial conditions that determine how the gas disperse in space and time (e.g., Yakovitch et al. 2015). In practice, given that the actual distance of an emission from the measurement vehicle, as well as actual emission height in a city, is unknown, accurate estimation of emission magnitude (choice of a suitable equation) is challenging. Emission estimates with different equations are discussed in the companion paper (Ueyama et al. 2025).

## 4 Data analysis

### 4.1 Leak Indication (LI) and Leak Point (LP)

In this study, we identified CH$_4$ enhancements in a similar manner to previous studies (von Fischer et al. 2017; Weller et al. 2019; Maazallahi et al. 2020; Defratyka et al. 2021; Wietzel and Schmidt 2023; Fernandez et al. 2022; Vogel et al. 2024; Ueyama et al. 2025). The baseline CH$_4$ mole fraction was defined as the 5th percentile of all the data obtained within ±2.5 min moving time window for individual data points. Subtracting the baseline from each measurement value allowed us to determine the CH$_4$ excess value. When the CH$_4$ excess exceeds 0.1 ppm, the data point is tagged as a Leak Indication (LI), as visualized in Fig. 4a. We then identified a Leak Point (LP) when 5 or more consecutive data points were tagged as LIs, which means that CH$_4$ excess values of > 0.1 ppm lasted for > 5 secs at LPs, as our measurement was at 1 Hz. The central locations (latitude and longitude) and the maximum excess CH$_4$ value among these consecutive data points were assigned to represent each LP. The

$C_2H_6$ mole fraction data were processed in the same manner to determine the $C_2H_6$ excess. To avoid possible influence from the vehicle exhaust, the data points with vehicle speed less than 1.5 km h$^{-1}$ was excluded.

It is noted that our threshold (0.1 ppm) is same as that in our companion paper (Ueyama et al. 2025), similar to that applied for measurements in German cities (5 % by Wietzel and Schmidt, 2023) and smaller than that for measurements in Paris, France (0.5 ppm by Defratyka et al. 2021) and Bucharest, Romania (0.2 ppm by Fernandez et al. 2022). In this regard, we found that the magnitude of $CH_4$ excess was generally smaller in the Tokyo area than in US and Europe cities, and the threshold (0.1 ppm) was appropriate for effective detection of LI as a $CH_4$ emitting location. For instance, a higher threshold of 0.2 ppm

would miss ~60% of LPs with smaller enhancements. In contrast, a lower threshold of 0.05 ppm would more than double the LP counts, but, due to comparable magnitudes of the observed baseline $CH_4$ variability, the LP detection would entail larger uncertainty. As shown in Fig. 5, the LPs with excess $CH_4 < 0.2$ ppm comprise 41 % of all the LPs found during the measurement campaign. About 93 % of the LPs fell in the range of excess $CH_4$ values of <1 ppm. On the other hand, we detected 28 LPs (5 %) exceeding excess $CH_4$ of 2 ppm with maximum of 19.8 ppm, of which 23 and 5 LPs were classified as biogenic and fossil

fuel sources (see Section 5).

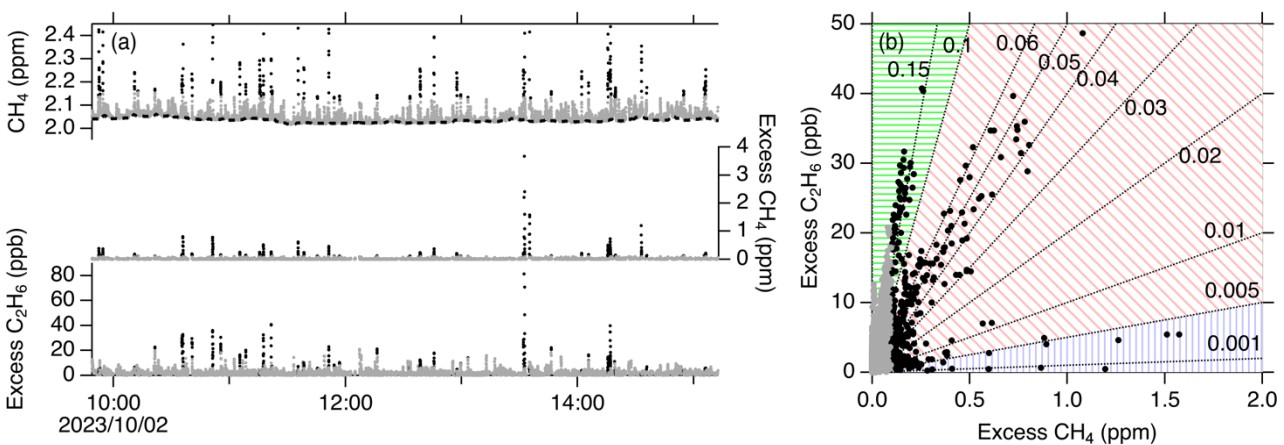

Figure 4: (a) An example of the Leak Indication (LI) methodology applied for the data obtained on 2 October 2023. The measured $CH_4$ mole fraction (top), excess $CH_4$ (middle) and excess $C_2H_6$ (bottom) on the driving vehicle are shown. The baseline $CH_4$ mole

fraction was calculated as the 5th percentiles of the measurement data within ±2.5 min moving time window (black dashed line) and the excess values were determined as deviations from the baseline. The excess $C_2H_6$ was calculated in the same manner. The data points identified as LI (the excess $CH_4$ of >0.1 ppm) are shown by black dots, while other data are coloured grey. (b) Scatterplots of excess $CH_4$ and $C_2H_6$ values for the data in panel (a). The data points are coloured in the same manner. Black dotted lines indicate slopes corresponding to different C2/C1 ratios (ppm ppm$^{-1}$) and each Leak Point (LP) was classified into biogenic (blue vertical-line

area), fossil fuel (red slant-line area) and combustion (green horizontal-line area) sources accordingly.

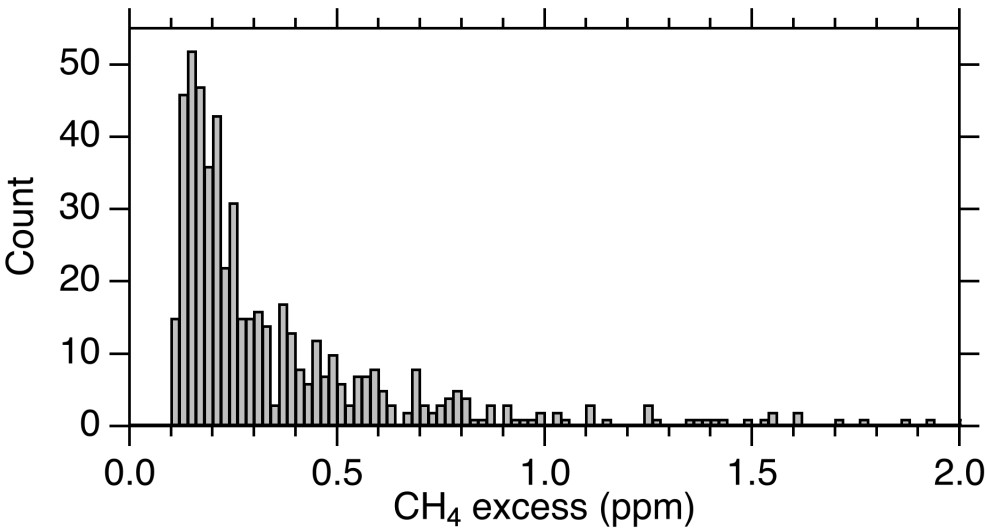

**Figure 5: Histogram of the excess CH$_4$ values of all the LPs observed during the measurement campaign. Note that data points with excess CH$_4$ of <0.1 ppm were excluded in the data processing, and that 5 % of LPs ($N$=28) exceeded excess CH$_4$ of 2 ppm (see text).**

**4.2 C$_2$H$_6$ to CH$_4$ ratio (C2/C1 ratio) and source classification**

As shown in Figs. 4a and 4b, we observed coincident increases of CH$_4$ and C$_2$H$_6$ mole fractions, and their enhancement ratio varied at different LPs. We determined C$_2$H$_6$-to-CH$_4$ enhancement ratios (C2/C1 ratio) for individual LPs. The C2/C1 ratio was used for source attribution of each LP according to the source classification by Fernandez et al. (2022); LPs with the C2/C1 ratio of <0.005, 0.005–0.1 and >0.1 are attributed to biogenic, fossil fuel and combustion sources, respectively. Note that the gas composition reported by the local gas company corresponds to the C2/C1 ratio of 0.063. For the measurement day shown in Fig. 4, we have clusters of the data categorized into biogenic (C2/C1<0.005), fossil fuel (C2/C1~0.05) and combustion (C2/C1~0.15).

**5 Result and Discussion**

Figure 6 presents LP locations identified during the measurement campaign. When classified into emission magnitude categories by previous studies (von Fischer et al. 2017; Fernandez et al. 2022; Ueyama et al. 2025), 531 (94 %) LPs were identified as low emissions (excess CH$_4$ of <1.6 ppm or <6 L min$^{-1}$ emission calculated by equation Weller et al. 2019), while 30 and 4 LPs were considered as medium (intermediate of low and high categories) and high (>7.6 ppm or >40 L min$^{-1}$) emissions, respectively. Among the 4 LPs of the high emission category, 3 of them were found near wastewater treatment plants and the rest was in a residential area (see below). The LPs were also grouped into three source categories (biogenic,

fossil fuel and combustion) according to the C2/C1 ratio as described in Section 4.2. Among 565 LPs identified during the whole measurement period, 300 LPs (53 %) were attributed to biogenic sources with little enhancement in $C_2H_6$, while 235 (42 %) and 30 (5 %) LPs were classified into fossil fuel and combustion origins, respectively. In this study, due to the very limited number of the combustion LPs, our analysis below mainly addresses biogenic and fossil fuel sources.

All the measurement data points were classified into different survey areas: 23 special wards of Tokyo Metropolis, Chiba and Kanagawa Prefectures. Maximum excess $CH_4$ values at LPs in different areas are shown in Fig. 7. Relatively high $CH_4$ excesses were found mainly in the special wards of Tokyo Metropolis, although some smaller enhancements were detected also in the bay areas of Chiba and Kanagawa Prefectures. In the bay areas of Chiba, 28 biogenic LPs, 38 fossil fuel LPs, and 7 combustion LPs were found. The biogenic and fossil fuel LPs with the highest $CH_4$ excesses of 1.44 and 2.27 ppm, respectively, were

located in the southern part of Keiyo Industrial Zone. In the bay area of Kanagawa (Keihin Industrial Zone), 3 biogenic LPs, 9 fossil fuel LPs, and 2 combustion LPs were found. The $CH_4$ excesses were at most 0.29 and 0.48 ppm for biogenic and fossil fuel LPs, respectively. In Tokyo Metropolis, we found large $CH_4$ enhancements such as biogenic LPs in Minato, Shinjuku, Sumida, Koto, Ota, Arakawa, Katsushika and Edogawa Wards as well as fossil fuel LPs in Shibuya and Toshima Wards. In these areas, many of biogenic LPs with large $CH_4$ enhancements (>1 ppm) were detected downwind of wastewater treatment

plants, indicating importance of the large facilities in $CH_4$ emissions in Greater Tokyo Area. Below we address characterization of overview from the campaign measurements with focus on quantification of areal emissions. Detailed analysis of emissions from individual known $CH_4$ sources, such as the landfill site and wastewater treatment plants will be presented in a separate follow-up study.

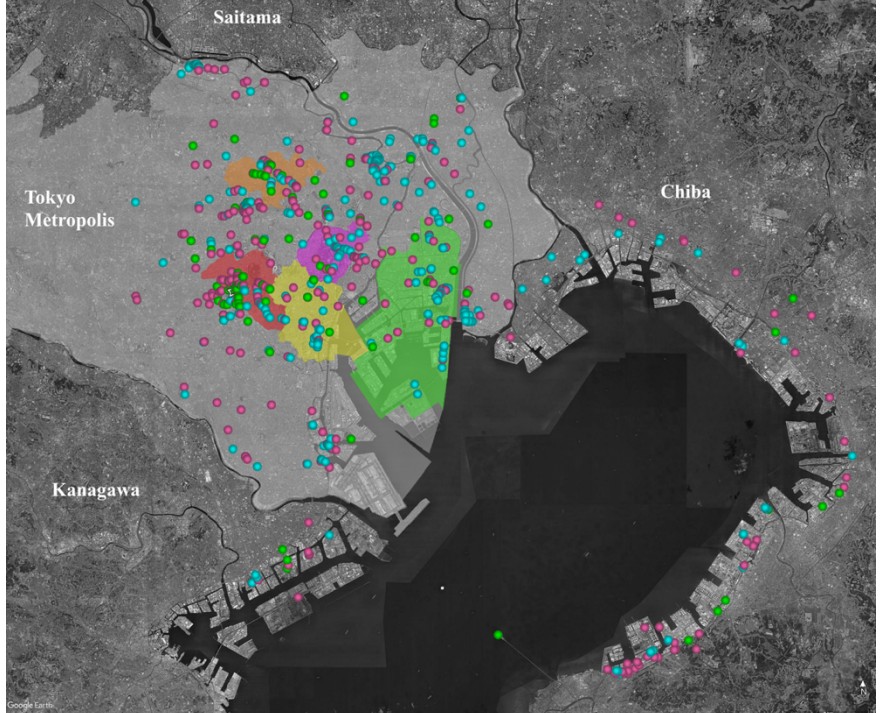

**Figure 6: Overview of the Leak Point (LP) locations for all the measurement days. The LP locations are indicated by circles and categorized by the observed C2/C1 ratios: blue for biogenic, pink for fossil fuel, and green for combustion sources. Areas are shaded same as Fig. 1. The map imagery was from © Google Earth.**

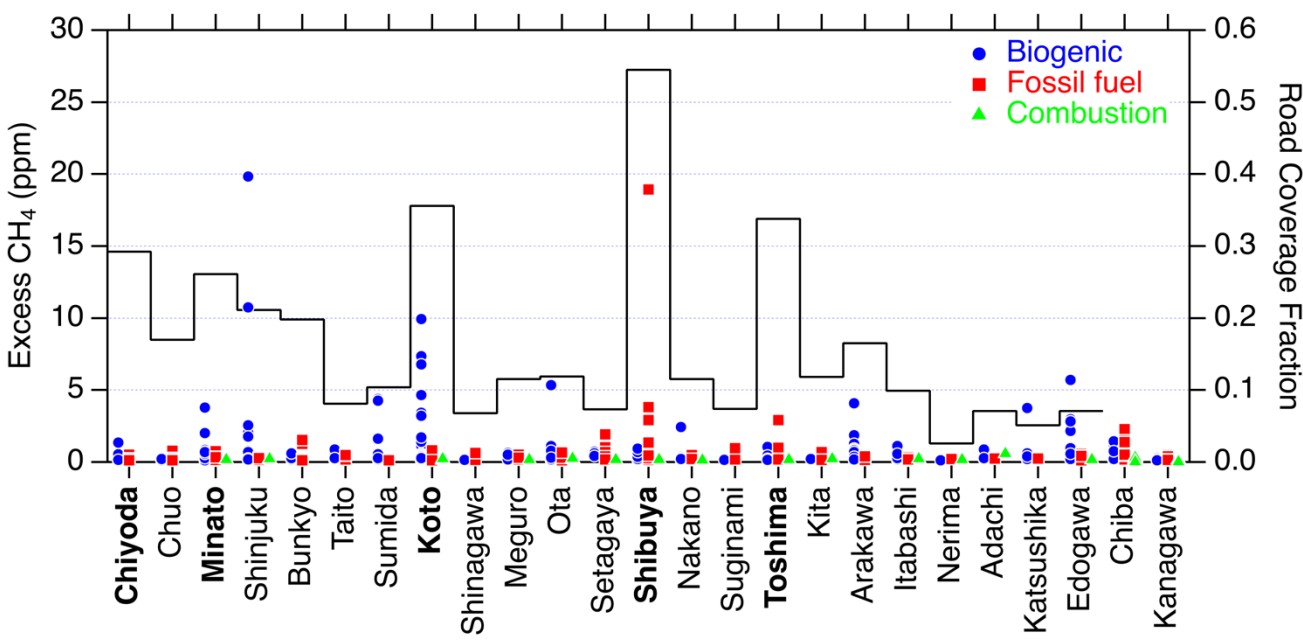

**Figure 7: Excess CH₄ values at the LP locations observed in the 23 special wards of Tokyo, Chiba and Kanagawa Prefectures grouped into the three source categories: biogenic (blue circles), fossil fuel (red squares) and combustion (green triangles) sources. Coverage of road distance in each area is also shown by black solid line (right axis), although not applicable to Chiba and Kanagawa Prefectures due to very limited road coverage. Five selected special wards are in bold.**

### 5.1 Characterization of LPs for different wards of Tokyo

Table 1 summarizes relevant statistics and the observed appearances of LPs for 5 selected wards. We note that the road distance covered by the present measurement campaign was only ~12 % of the total roads of Tokyo Metropolis. The coverage for each ward ranged from 2.6 % to 54.5 % (see Figure 7). As shown in Table 1, the best coverage was in Shibuya Ward, followed by Toshima, Koto, Chiyoda and Minato Wards (>25 %). Among 469 LPs found in Tokyo Metropolis, count of LPs ranged from

17 to 66 in these 5 wards, which correspond to LP densities from 0.16 to 0.47 km⁻¹. The LP density is count of LPs per travel distance and indicate average frequency of CH₄ enhancement encounters in a target city (Vogel et al. 2024; Ueyama et al. 2025). When grouped into source categories, biogenic and fossil fuel LP densities ranged from 0.07 to 0.35 km⁻¹ and from 0.08 to 0.20 km⁻¹, respectively. The average LP density in Tokyo Metropolis (0.33 km⁻¹) is comparable to that observed in Osaka (0.39 km⁻¹), the second largest city of Japan (Ueyama et al. 2025). When classified by source categories, the average

LP densities of biogenic and fossil fuel origins in Tokyo were 0.18 and 0.13 km$^{-1}$, respectively, in comparison to 0.13 and 0.24 km$^{-1}$ in Osaka. Vogel et al. (2024) presented comparisons of leak indication densities across cities in Europe and North America. Although not exactly compared due to differences in data processing, the fossil fuel LP density in Tokyo is apparently as large as those of most European cities with relatively small numbers of leak indications (roughly from Barcelona to Groningen in Figure 2 of Vogel et al. 2024). Note that we counted CH$_4$ enhancements of >0.1 ppm, which falls between thresholds used in the two classification methods of Vogel et al. (2024).

**Table 1: Measurement statistics and relevant information of the selected 5 wards of Tokyo Metropolis including road distances covered by the present measurements, areas and observed counts of LPs. The numbers in brackets in the column "Biogenic" show LP counts in the proximities of biogenic facilities (landfill and wastewater treatment plants).**

| Ward | Roads covered (km) | Total roads of the ward (km)[a] | Coverage fraction (%) | Area of the ward (km$^2$)[b] | LP count | | | | LP density (km$^{-1}$) |
|---|---|---|---|---|---|---|---|---|---|
| | | | | | Total | Biogenic (proximity of biogenic facilities) | Fossil fuel | Combustion | |
| Chiyoda | 51.29 | 175.57 | 29.2 | 11.66 | 17 | 9 (0) | 8 | 0 | 0.33 |
| Minato | 79.11 | 303.74 | 26.1 | 20.36 | 36 | 22 (12) | 12 | 2 | 0.46 |
| Koto | 125.32 | 395.91 | 35.6 | 42.99 | 66 | 49 (33) | 20 | 3 | 0.47 |
| Shibuya | 148.19 | 271.85 | 54.5 | 15.11 | 43 | 10 (0) | 30 | 3 | 0.29 |
| Toshima | 104.19 | 307.89 | 33.8 | 13.01 | 17 | 7 (0) | 8 | 2 | 0.16 |
| Total (23 wards) | 1426.37 | 11976.67 | 12.1 | 627.51 | 469 | 268 (149) | 186 | 21 | 0.33 |

[a]Bureau of Construction of Tokyo Metropolitan Government

[b]Tokyo Metropolitan Government

We analysed the observed frequency of the C2/C1 ratio in each ward. In many wards, as represented by Minato and Koto Wards, C2/C1 ratios close to zero (i.e., biogenic sources) were observed most frequently (Figure 8). In contrast, Shibuya Ward showed distinct characteristics of the C2/C1 ratio with the highest frequency at ~0.05, which is likely to correspond to fossil fuel sources. As a result, about three quarters of the LP counts in Shibuya Ward was attributed to the fossil fuel source (Table 1). Toshima and Chiyoda Wards also showed that about half of LPs was attributed to fossil fuel, although number of LPs were limited. The appearance of combustion LPs (C2C1 ratio of >0.1) was minor for all areas.

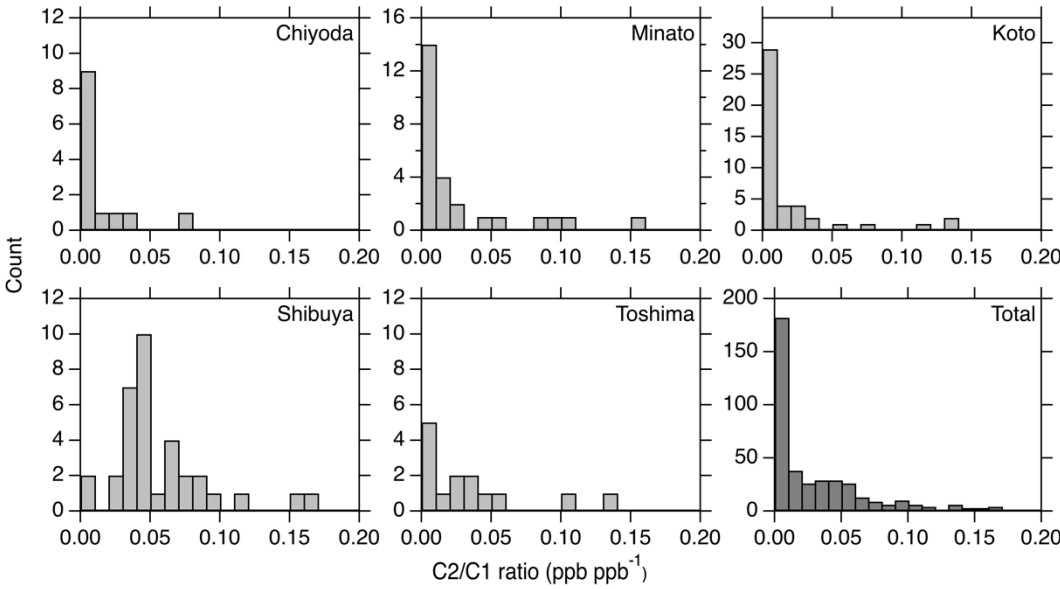


**Figure 8: Histograms of the C2/C1 ratios at individual LPs for the five selected wards.**

The different frequencies of the source attributions were plausibly associated with land uses and known $CH_4$ sources of the different wards. According to the report of Tokyo Metropolitan Government (Bureau of Urban Development 2023), Shibuya
and Toshima Wards are mainly covered by residential land (68.7 and 68.3 %, respectively), while the fraction is relatively low in Koto Ward (47.5 %). In addition, we note that our measurements prioritized coverage of residential areas for Shibuya Ward. The residential land of Koto Ward is occupied substantially by industrial areas (26.6 %), whereas the predominant coverage in other 4 wards are public, commercial, and residential spaces. It is also important to note that the present study surveyed streets near landfill sites (Tokyo Bay-side Landfill) and two wastewater treatment plants resided in Koto Ward, contributing
to large number of biogenic LPs in the area. Minato Ward also has a wastewater treatment plant surveyed. In Koto Ward, 2 and 31 LPs were found in the vicinities of landfill site and wastewater treatment plants, respectively, among 44 biogenic LPs in total (Table 1). In Minato Ward, 12 of 22 LPs were identified near the wastewater treatment plant. In contrast, our measurement did not survey any known biogenic $CH_4$ source facilities in Shibuya and Toshima Wards. It should be noted that the above LP counts (and its density) include those found at proximities of biogenic facilities as mentioned above. Other
biogenic LPs could be related to sewer networks and other water environment distributed in the survey areas. These facility and non-facility sources should be separately evaluated in a more sophisticated approach to estimating areal $CH_4$ emissions, but due to the limited data currently available, we do not distinguish these sources in the analysis below.

## 5.2 Emission estimates

To estimate areal $CH_4$ emissions, we employ empirical equations as referred in section 3 and previous studies. Also relevant is LP density ($km^{-1}$) defined as number of the observed LPs divided by the road distance covered in each ward (Table 1). Below we estimate $CH_4$ emissions from the 5 selected wards (Chiyoda, Minato, Koto, Shibuya and Toshima) only, because estimates for other wards are considered to be far from representative due to limited data. Using the average maximum $CH_4$ excesses in the area (Table 2), we calculated emission rates (L $min^{-1}$ or g $CH_4$ $min^{-1}$) according to Equation 4 and equations

suggested by previous studies. In this study, we present estimates using the equation by Weller et al. (2019) for compatibility with our companion paper (Ueyama et al. 2025) and previous studies (Maazallahi et al. 2020; Vogel et al. 2024). It is noted that, when Equation 4 (our control release experiment) was employed, the estimated emissions would be about 10 times larger than those using the Weller et al. (2019) equation, indicating that conversion from $CH_4$ excess to emission rate with an empirical equation is the considerable source of uncertainty. As described in Section 3, suitable choice of such an empirical

conversion equation requires consideration of important factors such as emission height and distance from the source in the target area. Use of Weller et al. (2019) equation assumes that most $CH_4$ emissions are represented by ground emissions occurring near the vehicle (roughly within 10 m). In contrast, Equation 4 (our control experiment) assumes emissions some tens m away at 5 m height. The former might approximate emissions underground sources (e.g., natural gas distribution pipes or urban sewer networks), whereas the latter downwind advection from large facilities or local restaurants at significant

distance (Ueyama et al. 2025). Since the survey areas have these mixtures, it is difficult to determine appropriateness of the equations for individual cases.

The areal flux was estimated by the following equation (Ueyama et al. 2025):

$$F = ER \times N \times RD/A \tag{5}$$

where $F$ is the areal flux (nmol $m^{-2}$ $s^{-1}$), $ER$ is the emission rate, $N$ is the number of LPs per travel distance ($km^{-1}$), $RD$ is the

total road distance (km), and $A$ is the total area of the ward ($km^2$). The estimated areal fluxes for the selected wards are tabulated in Table 2 and shown in Fig. 9a. As seen in Fig. 9a, reflecting significant counts of the measurement data downwind of the landfill site and wastewater treatment plants, the estimated fluxes for Minato and Koto Wards were contributed predominantly by biogenic sources. In contrast, emissions from residential areas in Shibuya and Toshima Wards were almost entirely attributed to fossil fuel sources.



**Table 2: The averages of the maximum CH₄ excess values and estimated and reported emissions for the 5 selected wards of Tokyo Metropolis. The uncertainties (range) of the average of the maximum CH₄ excess and emission estimates were calculated using a Bootstrap method. The reported emissions for the year 2021 were available at https://all62.jp/jigyo/ghg.html (in Japanese).**

| Ward | Average of maximum CH₄ excess (ppm) | Emission rate (L min⁻¹) | Areal flux (nmol m⁻² s⁻¹) | Areal emission (ktCH₄ yr⁻¹) | Reported emission (ktCH₄ yr⁻¹) |
|---|---|---|---|---|---|
| Chiyoda | 0.30±0.07 | 0.78 (0.55–1.01) | 2.89 (2.06–3.76) | 0.017 (0.012–0.022) | 0.04 |
| Minato | 0.46±0.11 | 1.31 (0.95–1.69) | 6.60 (4.78–8.52) | 0.068 (0.049–0.088) | 0.08 |
| Koto | 0.90±0.21 | 2.94 (2.11–3.81) | 10.3 (7.41–13.4) | 0.22 (0.16–0.29) | 0.12 |
| Shibuya | 0.97±0.43 | 3.23 (1.58–5.05) | 12.6 (6.15–19.6) | 0.096 (0.047–0.15) | 0.04 |
| Toshima | 0.46±0.16 | 1.28 (0.75–1.86) | 3.69 (2.16–5.36) | 0.024 (0.014–0.035) | 0.04 |

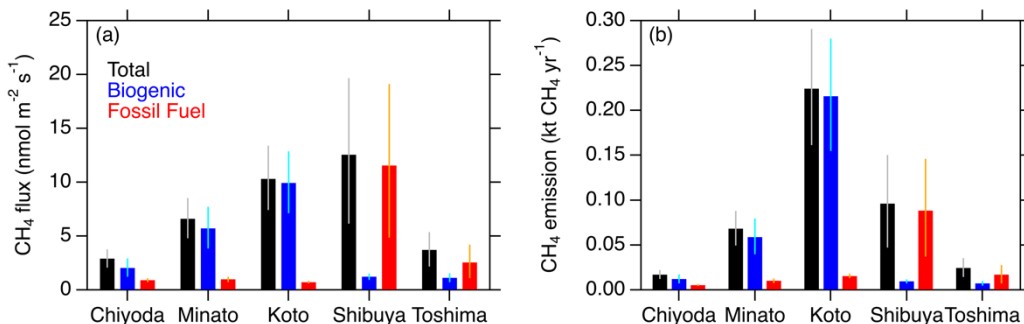


**Figure 9: (a) Total areal flux estimates (black) and those of biogenic (blue) and fossil fuel (red) sources for the 5 selected wards of Tokyo Metropolis. (b) Same as (a), but for upscaled CH₄ emission estimates. The error bars indicate uncertainty ranges calculated using a Bootstrap method.**

The areal fluxes can be converted to annual areal emissions (kt CH₄ yr⁻¹) by multiplying the total areas of the individual wards as shown in Table 2 and Fig. 9b. Based on our measurements, it was indicated that Koto Ward had the largest CH₄ emission of 0.22 kt CH₄ yr⁻¹, followed by Shibuya (0.096 kt CH₄ yr⁻¹), and Minato (0.068 kt CH₄ yr⁻¹) Wards. As discussed earlier, our estimates suggested that the areal emissions were predominantly of biogenic origin for Chiyoda, Minato and Koto Wards, whereas fossil fuel sources dominated the total emissions of Shibuya and Toshima Wards. In comparison to our estimates,

CH₄ emissions from these wards were reported to be 0.04, 0.08, 0.12, 0.04 and 0.04 kt CH₄ yr⁻¹ for respective wards (Table 2). Although these emission estimates are not considered to be conclusive due to large uncertainties, it is noteworthy that our biogenic emission estimates for the five selected wards were correlated with the reported emissions. This is apparently consistent with the reporting of the Tokyo Metropolitan Government that biogenic emissions account for most of the reported emissions according to the standard emission methodologies with nominal emission factors (Bureau of Environment of Tokyo

Metropolitan Government, 2024). However, for Shibuya and Toshima Wards, where our emission estimates attributed

primarily to fossil fuel sources, the discrepancies between the estimated and reported emissions were large. This implies that either the fossil fuel emissions observed in this study were not taken into account or emission factors for the relevant sectors employed for the report were smaller than actual.

Lastly, we stress that our emission estimates presented in this study are preliminary and that accurate evaluation of the emission
reporting by the local government is currently difficult. As described earlier, our survey covered only ~10 % of roads of the Tokyo Metropolis, and up to ~50 % at a maximum for the wards with relatively high measurement coverages. In addition, the survey prioritized measurements in the proximities of known $CH_4$ sources (mainly biogenic), by which the result may be biased towards biogenic emissions for the target wards with such known sources (Minato and Koto Wards). For residential areas where fossil fuel-related $CH_4$ enhancements were frequently found (Shibuya and Toshima Wards), the measurements were not
even in space, by which the current estimates may be underrepresentative for respective areas. We plan the second phase of mobile measurement campaign in Tokyo to increase the road coverage. It should be also noted that upscaled calculation from vehicle-based mobile measurements at ground level could underestimate citywide emissions, since such observations could miss $CH_4$ emission signals taking place at heights above street level (Ueyama et al. 2025). Moreover, as discussed earlier, the empirical conversion from the $CH_4$ excess to the emission rate (Section 3) is a source of large uncertainty, roughly a factor of
10. The equation will be better evaluated through comparison to the ongoing eddy covariance measurements at the site Yoyogi located in a residential area of Shibuya Ward (Sugawara et al. 2021), as performed by Ueyama et al. (2025). For preliminary comparison to Ueyama et al. (2025), $CH_4$ flux estimations for selected wards of Tokyo would fall within the range of variability found in various areas in Osaka and Sakai cities if the same conversion equation from excess $CH_4$ to emission rate was applied, which implies comparable magnitudes of urban $CH_4$ emissions in both megacity areas.


## 6 Concluding remarks

Based on the vehicle-based mobile atmospheric $CH_4$ and $C_2H_6$ measurement data newly obtained in September–October 2023, this study significantly improved our understanding of urban $CH_4$ emissions in Tokyo. It is clearly inferred that $CH_4$ emitting locations of fossil fuel origin, identified by the $C_2H_6$ to $CH_4$ enhancement ratio, are almost ubiquitous in residential areas,
which contrasts to the reporting of the Tokyo Metropolitan Government that account for 95 % of $CH_4$ emissions from waste sectors (biogenic $CH_4$). It is also shown that waste sectors, such as solid waste landfills and wastewater treatment plants, are consistently large $CH_4$ emitters, as high $CH_4$ enhancements were observed in downwind proximities. We also presented preliminary estimates of $CH_4$ emissions for 5 selected wards of Tokyo Metropolis where the measurement densities were relatively high. Although our preliminary estimates of areal $CH_4$ emissions are subject to large uncertainties, currently available
data imply that Tokyo's natural gas emissions are as low as those of Osaka (Japan's second largest megacity, Ueyama et al. 2025) and European cities of the lower-emission group (Vogel et al. 2024). However, the present study has raised further challenges for better understanding of urban $CH_4$ emissions in Tokyo. First, although 3-weeks of mobile measurements with

a driving distance of roughly 2000 km were performed, coverage of the public roads of Tokyo Metropolis was only ~10 %. Although the coverages exceeded 30 % in a few special wards of Tokyo, more measurements are needed to reduce possible

sampling biases in locating $CH_4$ emissions and attributing them to different sectors. Second, following earlier studies, this study also inferred a large uncertainty range originating from the empirical equation employed for conversion from $CH_4$ enhancement measurement to emission rate. To reduce uncertainties in estimating the upscaled emissions, usefulness of the equation in the measurement environment should be carefully evaluated, as examined in the companion paper (Ueyama et al. 2025). Even without such evaluation, mobile survey data can be utilized to support mitigation planning by identifying the

largest point source fugitive emissions and prioritizing for repair. With further work, and when uncertainties in measurement-based $CH_4$ emission estimates are sufficiently reduced with increased data, our methodologies may be able to support tracking of mitigation actions of local governments through identifying $CH_4$ emission locations, sectors and magnitude.

## Data availability

The dataset from the Tokyo mobile measurement campaign is available via Umezawa and Terao (2025).

## Author Contribution

TU, YT, and MU designed the study, conducted the measurement, and ensured data quality, with discussion of study goals with ML and JLF. TU analyzed the data with SK for analysis of the area allocation. TU wrote the manuscript with contributions

of all co-authors.

## Competing interests

The authors declare that they have no conflict of interest.

## Acknowledgements

This study was supported by the Environmental Defense Fund and Climate Change and Air Quality Research Program of

National Institute for Environmental Studies. The vehicle measurements were made by CLIMATEC, Inc. The control release experiment was conducted at the Methane Emission Facility of the JGC Corporation (JGC R&D Center in Oarai, Ibaraki, Japan). We thank Hideki Nara for helpful discussion about evaluation of the analyzer.

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
