# Peer review of "Measurement report: Mobile measurements to estimate urban methane emissions in Tokyo"

_EGUsphere, 2025_

## Author Comment (AC1)

We are grateful to the referee for thorough evaluation of our manuscript. Our responses to the comments are detailed below. Please note that *the referee's comment* and our responses are in different styles. The page and line numbers shown below correspond to those in the revised manuscript.

We note the following corrections: (1) Figure 4 (b) has been modified with color shades added according to the referee's suggestion. (2) The analyzer's name "MIRA ULTRA" at every place has been changed to "MIRA Ultra" to be consistent with the producer. The labels of Figure 3 have been corrected accordingly.

The manuscript presents a methane source characterization in the Tokyo area using mobile measurements of  $CH_4$  and  $C_2H_6$ . Although this technique is not novel, this is the first study of its kind in the Tokyo area. It is very interesting to see a comparison with other major cities, where most  $CH_4$  sources are of fossil origin, while in Tokyo biogenic sources seem to be more important. The manuscript is generally well written, and the instrument characterization is well detailed. However, I find the emission quantification method not sound, due to the issues related to the height of the emission release, the distance from the source and different wind conditions, which affect gas dispersion. I am not sure that, given the different meteorological and sampling conditions during the sampling campaigns, it is possible to quantify areal emissions by using the equation yielded during the control release experiment. I see this procedure as a general assessment of emissions rather than a tool for comparing the estimated fluxes to reported emissions, because the level of uncertainty associated with such estimates is simply too high. In the manuscript all these biases are described, but it is hard to reach a conclusion whether emission inventories underestimate methane sources or not, and I would give way less weight to the emission quantification part. Therefore, I suggest publication after addressing this issue and the following minor points:

We acknowledge that this comment indeed addresses the issue which has incompletely resolved in this study. It is why we presented detailed descriptions on possible sources of uncertainty in estimating emissions, and therefore we decided to avoid immediate discussion on possible under/overestimation of the inventories, as pointed out by the referee. The uncertainty in upscaling emissions using an empirical conversion from CH4 concentration enhancement to emission rate is a common problem in similar studies, and we continue to improve the methodologies by combining ongoing series of follow-up measurement studies in Tokyo and Osaka (vehicle and stationary measurements). Nevertheless, in the present manuscript, we consider that it is worth presenting our preliminary estimates as our current

best knowledge. In particular, we hope to highlight implication from our emission estimates that urban fossil fuel sources with magnitude comparable to biogenic sources are not taken account, which is qualitatively valid even with the large uncertainty in magnitude of the estimated emissions.

Given this comment, we revisited every relevant place in the manuscript to minimize possible misleading wordings. The following corrections have been made; in Section 5.2, (1) One sentence ("Our estimates are therefore may appear comparable to the reported emissions.") has been left out and the following sentence has been modified to "Although these emission estimates are not considered to conclusive due to large uncertainties, it is noteworthy that…" (P17 L411) (2) The first sentence in the last paragraph of Section 5.2 has been modified to "Lastly, we stress that our emission estimates presented in this study are preliminary and that accurate evaluation of the emission reporting by the local government is currently difficult." (P18 L419)

Line 117: the repeatability of.. Corrected.

**Line 128: explain here why you chose the 1 ppm threshold. You explained that later, but I feel that we need more explanation at this stage**

The "> 1 ppm" here did not mean any threshold. It was intended to indicate that a possible bias of  $\sim$ 25 ppb (0.025 ppm) at the CH4 mole fraction level of  $\sim$ 5 ppm was small in comparison to the observed variability with magnitude of >1 ppm, which means that possible corresponding correction would not change analysis results. We have rephrased the sentence as follows.

P5 L133: "As this possible bias (~0.025 ppm) is relatively small in comparison to the observed variability (>1 ppm excess values with respect to the baseline), we apply no corrections for CH4 measurements during the Tokyo mobile campaign measurements."

**130: why do the C2H6 values by MIRA Ultra vary so much?**

Although we cannot identify a single cause, it is plausibly to do with reproducibility of quantification of absorption specific to the low concentration  $C_2H_6$  (i.e., <1 ppb at background). Uncertainty at initial determination of the  $C_2H_6$  absorption spectrum when turned on may cause day-by-day biases.

**137: "The mole fractions here are uncorrected as..". This sentence is not clear, please rephrase.**

We have corrected the sentence as follows.

P6 L144: "The mole fractions here are reported values by the instrument and applied no corrections."

**Figure 165: change the y axis label. Should this be "difference from the nominal value (ppm)..?". Please clarify**

To keep the axis labels short, we hope to keep them same, but we have added the following sentence in the figure caption.

Figure 2 caption: "Note that CH4 and C2H6 mole fractions are plotted as differences from nominal values."

**Line 244-249: I would move these sentences to the result section**

Corrected according to the suggestion.

**Figure 4 (b): could you change the colors of red lines according to the source category (e.g. fossil, biogenic and combustion)?**

We have corrected the figure as attached below so that readers can visually understand the source categories. The figure caption was modified accordingly.

**Line 270-272: Move to the result section**

Corrected according to the suggestion.

**Line 307: I would explain the concept of LP density here**

We have inserted the following sentence.

P13 L320: "The LP density is count of LPs per travel distance and indicate average frequency of CH4 enhancement encounters in a target city (Vogel et al. 2024; Ueyama et al. 2025)."

**Table 2: I am not very convinced about reporting these emission estimates, see my previous comment**

We appreciate the referee's criticism. As explained above, we still consider that the emission calculation is worth presenting at least to explain our qualitative conclusion. We have revised our descriptions so that readers do not misunderstand that our calculation is quantitatively conclusive.

---

## Author Comment (AC2)

We are grateful to the referee for thorough evaluation of our manuscript. Our responses to the comments are detailed below. Please note that *the referee's comment* and our responses are in different styles. The page and line numbers shown below correspond to those in the revised manuscript.

We note the following corrections: (1) Figure 4 (b) has been modified with color shades added according to another referee's suggestion. (2) The analyzer's name "MIRA ULTRA" at every place has been changed to "MIRA Ultra" to be consistent with the producer. The labels of Figure 3 have been corrected accordingly.

This study provides valuable insights into urban methane emissions in Tokyo, employing mobile vehicle-based measurements of  $CH_4$  and  $C_2H_6$ . The work is timely and relevant for improving city-level emission inventories, especially given discrepancies between bottom-up (Tokyo Metropolitan Government, EDGAR) and observation-based estimates. The integration of a controlled release experiment is a strength, although it also highlights significant uncertainties in emission quantification.

The manuscript makes a novel and policy-relevant contribution to understanding urban methane emissions. I recommend minor revisions to address the issues of emission quantification uncertainty, spatial representativeness, and inventory implications. With these improvements, the study will provide a strong addition to the literature on methane emissions in megacities.

Overall, I find this paper suitable for publication after addressing several concerns outlined below.

We again thank the referee for constructive evaluation of our manuscript. Please see our responses below.

**Minor Comments**

L20-24: The abstract's conclusion is vague. It mentions an inconsistency with local government reports for residential areas but doesn't specify how. To enhance the impact, clearly state whether the measured emissions were higher or lower than the reported values. This highlights the need for improved accounting in specific sectors and underscores the value of direct-measurement research. The abstract effectively introduces the scope and importance of the study but lacks quantified uncertainty. Including error ranges (e.g., ± values for emission estimates) or highlighting explicit policy implications would improve clarity for broader audiences.

We appreciate this comment which pinpoints importance of this study. It is unfortunately difficult for the moment to clearly state possible under/overestimation of local government reporting due to the large uncertainties in our emission estimates. Our current best arguments from the preliminary emission calculations are (1) the estimates are well correlated with the reporting for the areas with waste-sector facilities (Chiyoda, Minato and Koto Wards), showing actual key roles of emissions from the facilities, and (2) fossil fuel emissions in the residential areas (Shibuya and Toshima Wards) are comparable in magnitude to those in the areas with the waste facilities, but they are not accounted in the reporting. We have revised the last sentence of the abstract as follows:

P1 L23: "In the areas with biogenic facilities (landfill and wastewater treatment plants), our emission estimates are well correlated with local government reporting, indicating actual key contributions of the waste-sector facilities. On the other hand, in the residential areas, CH4 emissions were predominantly of fossil-fuel origin, with a magnitude comparable to the area with waste facilities. However, such fossil-fuel emissions are not accounted for in local

government reporting. This result highlights the need for improved accounting of urban fossil fuel-related emissions."

Line 14-16 The phrase "conversion is not straightforward" is vague. Suggest clarifying what aspects are most uncertain (e.g., dispersion variability, background definition, or calibration transferability).

We have corrected the sentence and added a new sentence as follows:

P1 L14: "The empirical equation derived from the experiment was significantly different from those reported by previous studies, suggesting the limitation of such enhancement-to-emission rate conversion, which is a source of large uncertainty in estimating urban CH4 emissions based on street-level measurements. The uncertainty stems from different experiment settings and underlying assumptions (e.g., source distance and height) which do not always represent actual urban measurement environments."

L64-70: I agree with that "Discrepancy between the local government reporting and the global data commonly used in the atmospheric science community". The discussion of the discrepancy between local government reporting and global datasets is important and well-motivated. While the year-to-year differences in reported values may not be dramatic, the comparison currently refers to different years (2021 vs. 2023). I recommend explicitly noting this difference in the manuscript, as it will help readers interpret the magnitude of discrepancies more clearly.

We appreciate the referee for the important comment. Since Bureau of Environment of Tokyo Metropolitan Government updated the greenhouse gas emission reporting in the meantime, we have shown the emission for 2023, by which comparison to EDGAR is now made for the same year. The update is small as in general CH4 emission in the Tokyo reporting gradually decreases. In addition, we have modified the relevant sentence as follows:

P3 L73: Given the general small decreasing trend of CH4 emissions of Tokyo (e.g., 0.5 % yr-1 in the Tokyo Metropolitan Government reporting), comparison of these datasets indicates discrepancy in magnitude and attribution of CH4 sources in Tokyo between the local government reporting and the global data commonly used in the atmospheric science community, highlighting the importance to improve activity-based CH4 emission datasets for Tokyo.

L93-95: The study covers ~2,000 km of road network (~10% of Tokyo), with higher coverage in selected wards, particularly near waste facilities. This uneven sampling may affect representativeness and source attribution. A brief note on this limitation and its implications for generalizing results would improve clarity.

We acknowledge the possible sampling bias, which is discussed in Section 5.2. For conciseness, we do not add explanation here.

L180-220: The Control Release experiment is a valuable addition. However, since it was conducted under specific wind conditions, its generalizability is somewhat limited. It would strengthen the manuscript if the authors briefly acknowledge this limitation and note that further experiments under varied meteorological conditions could be beneficial.

To mimic actual on-street measurements, the control release experiment was conducted downwind of the point source with wind speed of <3 m s $^{-1}$ , as described in the manuscript. The meteorological condition agrees to predominant summertime weather in the area, and more importantly, the wind speed was well within the variability in the Tokyo area (Japan Meteorological Agency, 2025). Therefore, we consider that the result is well representative and useful for the present study, although we acknowledge that differences from actual

measurements were made under to some degree various meteorological conditions. We have added the following sentence in the revised manuscript:

P8 L201: "These weather conditions agree to those typically observed at the site and in Tokyo (Japan Meteorological Agency, 2025)."

Japan Meteorological Agency, Tables of Monthly Climate Statistics: <a href="https://www.data.jma.go.jp/stats/data/en/smp">https://www.data.jma.go.jp/stats/data/en/smp</a>, last access: 29 October 2025.

L231-233: The choice of a 0.1 ppm threshold for defining LPs is central but appears somewhat arbitrary. While Tokyo's relatively low  $CH_4$  enhancements justify a lower threshold, a statistical validation (e.g., false positive/negative analysis) would provide stronger support.

We have added the following sentences in Section 4.1:

P10 L249: "For instance, a higher threshold of 0.2 ppm would miss  $\sim$ 60% of LPs with smaller enhancements. In contrast, a lower threshold of 0.05 ppm would more than double the LP counts, but, due to comparable magnitudes of the observed baseline CH4 variability, the LP detection would entail larger uncertainty."

L264-272 and overall results: The use of C2/C1 thresholds (<0.005 biogenic, 0.005–0.1 fossil fuel, >0.1 combustion) is appropriate, but the combustion category is underrepresented (only ~5% of LPs, ~30 samples). Given the small sample size, confidence in combustion attribution is limited. In dense residential areas with potential mixed sources, interpretation could be more complex. It would be useful to briefly acknowledge this limitation and, if appropriate, mention complementary methods (e.g., isotopes, co-tracers)

We agree to the referee. The following sentence has been added. For conciseness of this manuscript, we avoid additional discussion about possible complementary tracers. P12 L288: "In this study, due to the very limited number of the combustion LPs, our analysis below mainly addresses biogenic and fossil fuel sources."